# On Adaptive Knowledge Distillation with Generalized KL-Divergence Loss for Ranking Model Refinement

Anonymous

## ABSTRACT

Knowledge distillation is useful in training a neural document rank-
ing model by employing a teacher to guide model refinement. As a
teacher may not perform well in all cases, over-calibration between
the student and teacher models can make training less effective.
This paper studies a generalized KL divergence loss in a weighted
form for refining ranking models in searching text documents, and
examines its formal properties in balancing knowledge distillation
in adaption to the relative performance of the teacher and student
models. This loss differentiates the role of positive and negative
documents for a training query, and allows a student model to take
a conservative or deviate approach in imitating the teacher's behav-
ior when the teacher model is worse than the student model. This
paper presents a detailed theoretical analysis with experiments on
the behavior and usefulness of this generalized loss.

**ACM Reference Format:**
Anonymous. 2024. On Adaptive Knowledge Distillation with Generalized
KL-Divergence Loss for Ranking Model Refinement. In *Proceedings of ACM
Conference (Conference'17)*. ACM, New York, NY, USA, 10 pages. https:
//doi.org/10.1145/nnnnnnn.nnnnnnn

## 1 INTRODUCTION

Large-scale search systems for text documents typically employ
multi-stage ranking in practice. The first retrieval stage extracts
top candidate documents matching a query from a large search
index with a fast and relatively efficient ranking method. The sec-
ond stage or a later stage uses a more complex machine learn-
ing algorithm to re-rank top results thoroughly. Recent sparse
retriever studies exploit learned neural representations DeepIm-
pact [28], uniCOIL [12, 22] and SPLADE [7, 9]. An alternative
method is dense retrieval which uses a dual encoder architecture
with single-vector [33, 48], multi-vector document representations
(e.g. [17, 34]).

To boost the relevance of these models, knowledge distillation [14]
is critical during training to transfer knowledge from a powerful
teacher model through behavior imitation [11, 15, 23]. KL diver-
gence is a popular training loss for knowledge distillation in docu-
ment ranking [23, 33, 34, 42, 43].

One drawback of KL-divergence loss for document ranking is
that it does not exploit characteristics of contrastive learning in
ranking model refinement because it does not differentiate positive
and negative documents for a training query. As a result, it over-
calibrates between the student and teacher models with a tight
distribution matching in every document without prioritization

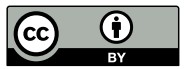

*Conference'17, July 2017, Washington, DC, USA*
© 2024 Association for Computing Machinery.
ACM ISBN 978-x-xxxx-xxxx-x/YY/MM...$15.00
https://doi.org/10.1145/nnnnnnn.nnnnnnn

even when the teacher performs worse than the student model.
The previous work has used the weighted sum of a contrastive loss
such as log-likelihood with KL divergence as a regularization to
reduce over-fitting, and a recent BKL study [45] improves this by
regularizing KL divergence with an entropy and L1-norm loss. As
discussed in Section 2, this BKL regularization can behave incor-
rectly in three significant case regions when a teacher is better than
and is worse than a student model for a training query.

To address the aforementioned weakness, the contribution of this
paper is a generalized KL-divergence loss formula called weighted
KL divergence (WKL) with a detailed analysis of its theoretical
properties. Instead of following the regularization approach, this
generalized loss guides knowledge distillation adaptively in ranking
model refinement by differentiating the role of positive and negative
documents and prioritizing the alignment of a student model and
a teacher model for effectively separating positive and negative
documents. This paper provides a loss lower bound analysis and a
relative gradient contribution study to characterize the behavior
of WKL during model training, compared to KL divergence. Our
analysis shows that this generalized loss can dynamically assess
the relative performance of the teacher and student model in each
training query, and adaptively adjust the imitating behavior of
the student model. so that the teacher model is followed when
it performs better than the student model, and is conservatively
followed or not followed at all when this teacher performs worse
than the student model.

This paper also provides experimental evidence that WKL out-
performs other loss options and examines sensitivities of WKL
parameters when refining three student models including SPLADE
sparse retrieval, ColBERT ranking with a multi-vector representa-
tion [34], and a single-vector SimLM dense retriever [42].

## 2 BACKGROUND AND RELATED WORK

**Problem definition.** We follow the notation used in [45]. Given
query $Q$, document search on a collection of $N$ text documents
(i.e., $\mathcal{D} = \{d_i\}_{i=1}^{N}$) finds top $k$ results with a ranking mainly based
on their query-document similarity. For training a retriever or re-
ranker, contrastive learning is widely used. Let $\mathcal{D}^+$ be the subset of
all positive documents, and $\mathcal{D}^-$ be a subset containing all negative
documents for query $Q$. We assume that in a training dataset, all
positive documents are ranked equally. That is true for the MS
MARCO passage dataset where there are only binary labels.

The top one probability distribution over these documents is:

$$P(d_i|Q, \mathcal{D}^+, \mathcal{D}^-, \Theta) = \frac{\exp(S(Q, d_i, \Theta))}{\sum_{j=1}^{N} \exp(S(Q, d_j, \Theta))}$$

where $\Theta$ is the vector of neural parameters involved. $S(Q, d_i, \Theta)$
is a scoring function that captures the semantic similarity of a
document with a query. For the simplicity of presentation when
no confusion is caused, we will not list $\Theta$ and $Q$ explicitly in each

symbol below and the loss function is specified for each query $Q$ based on parameters $\Theta$ under the training documents $\mathcal{D}^+$ and $\mathcal{D}^-$. Knowledge distillation is a training methodology that guides the refinement of a neural student model using a teacher model. Let $p_i$ or $q_i$ denote $P(d_i|Q, \mathcal{D}^+, \mathcal{D}^-, \Theta)$ where $p_i$ and $q_i$ refer to the teacher's and student's predictions, respectively.

To train a ranking model, the standard loss function includes the negative log-likelihood or its variation: $-\sum_{d_j \in \mathcal{D}^+} \log q_j$. KL-divergence defined below is a popular choice for knowledge distillation as seen in recent ranking studies [33, 34, 38, 42, 49].

$$L_{KL} = \sum_{d_i \in \mathcal{D}^+ \cup \mathcal{D}^-} p_i \ln \frac{p_i}{q_i} \tag{1}$$

where $p_i$ and $q_i$ refer to the teacher and student's top one probability for instance $d_i$ in $\mathcal{D}^+$ or $\mathcal{D}^-$, respectively. KL-divergence measures the distance between teacher's and student's distributions. It is known that the lower bound of KL-divergence loss is 0 and this is achieved when $\forall d_i, p_i = q_i$.

**Related retrieval methods.** Large-scale search systems for text documents typically employ multi-stage ranking in practice. The first stage retriever aims to fetch top $k$ documents using a fast and relatively simple method. There are two categories of retrieval techniques in deriving a document and query representation. One category of document retrieval is lexical sparse retrieval models, such as BM25, which take advantage of fast inverted index implementations on CPUs. This method gains its popularity recently due to the advancement of learned sparse representations that derive token weights from a BERT-based neural model [6, 9, 12, 22, 28, 37].

Dense retrieval is an alternative approach for first-stage search with a dual encoder architecture (e.g. [10, 44]). Distillation is shown to be effective for dense retrieval training and KL-divergence loss is a popular choice in recent studies, such as RocketQAv2 [33], SimLM [42] and RetroMAE [43], AR2 [49], and UnifiedR [38].

**Re-ranking and multi-vector representations.** The second or later stage of search can employ a more complex re-ranker to re-evaluate the top $k$ documents fetched by an earlier stage. There is a possibility to use a single-vector dense retrieval model for re-ranking. As pointed out in recent studies [21, 35, 40], single-vector dense models can struggle in handling out-of-domain datasets where training data is limited (including zero-shot retrieval), and in answering entity-centric questions. As a remedy, multi-vector representations including ColBERT and its new enhancements [20, 21, 27, 31] have been proposed to improve the model expressiveness by capturing fine-grained token-level information.

**Listwise losses.** A listwise loss design that considers the impact of relative rank positions of matched documents for a query has been shown to be useful in learning-to-rank and aligning such a loss with a targeted ranking metric approximately such as NDCG is ideal [25, 41]. Since neural information retrieval typically requires a large number of training examples to be effective, and training data such as MS MARCO only contains few labeled positive documents and sampled negative documents on a relatively large scale, it is more important to separate positive and negative documents properly for a query-specific loss. This motivates our design. The previous work has considered the relevance gain by swapping two documents in a listwise loss, e.g. LambdaMART [1]. CL-DRD [47] uses a listwise loss based on rank position. Weighting training

instances is studied in the focal loss for visual object classification [24], and such a loss is not designed for knowledge distillation. Nevertheless, our work is influenced by the above studies.

**Regularization of knowledge distillation with a contrastive loss**. A key weakness of knowledge distillation with KL divergence loss for document ranking is that a teacher model may not perform well in all cases and adaptive deprioritization is needed. A common approach to balance knowledge distillation is to combine the KL divergence loss with a contrastive rank loss such as the log-likelihood using a weighted sum as a regularization, defined as:

$$L_{KLL} = \sum_{d_i \in D^+ \cup D^-} p_i \ln \frac{p_i}{q_i} - \lambda \sum_{d_i \in D^+} \log q_i. \tag{2}$$

The above loss is not adaptive to the relative performance of a teacher model and a student model. An improvement called BKL [45] combines the negative entropy component of positive documents and the L1-norm expression of negative documents for a given query to balance knowledge distillation.

## 3 LOSS DESIGN AND ANALYSIS

### 3.1 Design considerations

Our goal of loss design optimization is to control the imitation of the teacher's rank scoring when refining a student model based on each training query so that when the student mimics when the teacher is better and it should restrain distillation or deviate when the teacher is worse. This can be analyzed by examining the gradient contribution of each document for parameter update during SGD-based training compared to KL divergence, as illustrated in Figure 1, and a desired loss should follow the gradient update direction of KL divergence loss when the teacher model performs better than the student model for a training query. When this teacher performs worse, this targeted loss should deviate in an opposite update direction or at least restrain the update size cautiously even in the same update direction.

The weakness of BKL [45] is that its formula over-corrects the behavior of KL divergence and fails to meet the above objective in three significant case regions. As shown in Section 4, when a teacher's model performs much better than a student in ranking a negative example, BKL's regularization formula unintentionally lets the student model deviate from the teacher's ranking score in a wrong learning direction. It also fails in some cases to follow aggressively even when the teacher model performs worse for a positive document.

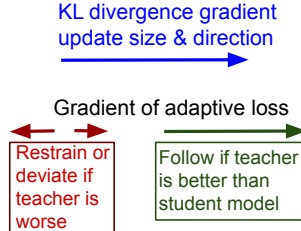

**Figure 1: Loss design goal: Adaptive control of student model learning from teacher compared to the KL divergence loss**

To meet the above expectation and goal illustrated in Figure 1, and avoid the misbehavior in both BKL and KLL, our approach described below does not take the regularization approach of BKL and KLL. Instead we directly generalize the KL divergence loss with an easy-to-implement weighting formula described in Section 3.2. In generalizing the KL divergence loss, our main strategy is to explicitly prioritize the separation of positive and negative documents for each query through a weight adjustment by down-weighting positive documents ranked high on the top positions, and negative documents ranked low at the bottom positions by a student model. This is illustrated in Figure 2 where documents are sorted from left to right in a non-decreasing order of their student rank score.

It is not obvious that the goal illustrated in Figure 1 could be accomplished by the above simple strategies illustrated in Figure 2 for ranking. Through a gradient contribution analysis, Section 4 analytically reveals that it is true under the generalized KL divergence loss described below. Namely, it allows learning from a teacher become better behaved in adaptation to the relative performance of the teacher model and the student model, and can restrain imitation when this teacher performs worse than the student.

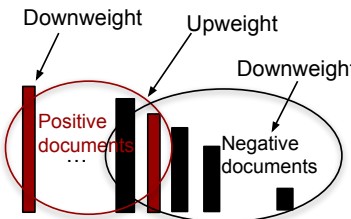

**Figure 2: Prioritize separation of positives and negatives**

## 3.2 Generalized KL-Divergence

This generalized KL-divergence loss in a weighted format (WKL) is defined as follows:

$$L_{WKL} = \sum_{d_j \in \mathcal{D}^+} (1-q_j)^{\gamma_1} p_j \ln \frac{p_j}{q_j} + \sum_{d_i \in \mathcal{D}^-} (q_i)^{\gamma_{2,i}} p_i \ln \frac{p_i}{q_i}.$$

The weight for each divergence term $p_i \log \frac{p_i}{q_i}$ corresponds to the importance to align the student's scoring of such a document with the teacher's model. For a positive document $d_j$, the goal is to have $q_j$ as large as possible towards 1, and thus we use $(1-q_j)^{\gamma}_1$ as the weight. Here $\gamma_1$ is a fixed hyperparameter controlling the scale of weight in the exponent. We require $\gamma_1 \geq 0$. For a negative document $d_i$, the goal is to have $q_i$ as small as possible towards 0, and thus we use $(q_i)^{\gamma_{2,i}}$ as the weight. We require either $\gamma_{2,i} > 0$ for all negative documents or $\gamma_{2,i} = 0$ for all negative documents. For two negative documents $d_i$ and $d_j$ where $q_i \geq q_j$, we require $\gamma_{2,i} \leq \gamma_{2,j}$.

Notice that KL divergence loss is a special form of WKL when setting all control parameters as zero ($\gamma_1 = \gamma_{2,i} = 0$). WKL weights the divergence loss contribution from positive documents and negative documents differently. We explain how the above design matches the design consideration illustrated in Figure 2.

- Given two positive documents $d_i$ and $d_j$, if $q_i \geq q_j$, then $(1 - q_i)^{\gamma_1} \leq (1 - q_j)^{\gamma_1}$. Thus a low-scoring positive document is weighted more than a high-scoring positive document. When such a document is ranked close to negative documents, or even below some negative documents, that results in a poor boundary separation of positive and negative documents. Thus the alignment with the teacher's model for such a positive document should be prioritized.
- Among negative documents, if $q_i \leq q_j$, requiring $\gamma_{2,i} \leq \gamma_{2,j}$ implies that $(q_i)^{\gamma_{2,i}} \geq (q_i)^{\gamma_{2,j}} \geq (q_i)^{\gamma_{2,j}}$. High-scoring negative documents are weighted more and low-scoring negative documents have a reduced priority to follow what the teacher does. When the score of a negative example in a student model is high and is getting closer or exceeds some of the positive examples, the positive and negative document regions would overlap as shown in Figure 2 and then that is a high-priority case to address.

## 3.3 Loss minimization and its bound

The result below shows that the WKL loss has a constant lower bound, and thus training that minimizes such a loss has a boundary to hit. If a loss function has no lower bound, training would not converge. Note that $p_i$ values from the teacher's model are constant.

**Theorem** 1. *Loss minimization. When $\gamma_1 \geq 1$ or $\gamma_1 = 0$,*

$$L_{WKL} \geq \sum_{d_j \in \mathcal{D}^+} p_j \ln \frac{p_j}{q_j} - \sum_{d_i \in D^-} p_i q_i^{\gamma_{2,i}} \ln q_i \\ + \frac{\gamma_1}{\log e} \sum_{d_j \in \mathcal{D}^+} q_j \log q_j + \sum_{d_i \in \mathcal{D}^-} p_i \ln p_i. \quad (3)$$

*When $0 < \gamma_1 < 1$,*

$$L_{WKL} \geq r_1 \sum_{d_j \in \mathcal{D}^+} p_j \ln \frac{p_j}{q_j} - \sum_{d_i \in \mathcal{D}^-} p_i q_i^{\gamma_{2,i}} \ln q_i + \frac{\gamma_1}{\log e} \sum_{d_j \in \mathcal{D}^+} q_j \log q_j \\ + (1 - \gamma_1) \sum_{d_j \in \mathcal{D}^+} p_j \ln p_j + \sum_{d_i \in \mathcal{D}^-} p_i \ln p_i. \quad (4)$$

PROOF. When $\gamma_1 \geq 1$, we follow Bernoulli's inequality $(1 - q_i)^{r_1} \geq 1 - r_1 q_i$, given $0 \leq q_j \leq 1$. Since $\ln p_i \leq 0$ and $\ln q_i \leq 0$,

$$L_{WKL} = \sum_{d_j \in \mathcal{D}^+} (1 - q_j)^{\gamma_1} p_j \ln p_j + \sum_{d_i \in \mathcal{D}^-} q_i^{\gamma_{2,i}} p_i \ln p_i \\ - \sum_{d_j \in \mathcal{D}^+} (1 - q_j)^{\gamma_1} p_j \ln q_j - \sum_{d_i \in \mathcal{D}^-} q_i^{\gamma_{2,i}} p_i \ln q_i \\ \geq \sum_{d_j \in \mathcal{D}^+} p_j \ln p_j + \sum_{d_i \in \mathcal{D}^-} p_i \ln p_i \\ - \sum_{d_j \in \mathcal{D}^+} (1 - q_j \gamma_1) p_j \ln q_j - \sum_{d_i \in \mathcal{D}^-} q_i^{\gamma_{2,i}} p_i \ln q_i \\ \geq \sum_{d_j \in \mathcal{D}^+} p_j \ln \frac{p_j}{q_j} - \sum_{d_i \in D^-} p_i q_i^{\gamma_{2,i}} \ln q_i \\ + \frac{\gamma_1}{\log e} \sum_{d_j \in \mathcal{D}^+} q_j \log q_j + \sum_{d_i \in \mathcal{D}^-} p_i \ln p_i$$

When $\gamma_1 = 0$,

$$L_{WKL} \geq \sum_{d_j \in \mathcal{D}^+} p_j \ln \frac{p_j}{q_j} - \sum_{d_i \in \mathcal{D}^-} q_i^{\gamma_{2,i}} p_i \ln q_i$$
$$+ \sum_{d_i \in \mathcal{D}^-} p_i \ln p_i.$$

When $0 < \gamma_1 < 1$, we first show that $(1 - x)^{\gamma_1} \geq \gamma_1(1 - x)$ with $x \in [0, 1]$. It is true when $x = 0$ and $x = 1$. For $x \in (0, 1)$, let function $f(x) = (1 - x)^{\gamma_1} + \gamma_1 x$. Then $f'(x) = -r_1(1 - x)^{\gamma_1 - 1} + \gamma_1 < 0$. Then $f(x)$ is monotonically decreasing and $f(x) > f(1)$ for $x \in (0, 1)$, which leads to $(1 - x)^{\gamma_1} > \gamma_1(1 - x)$. We apply this inequality for $x = q_j$ for a positive document below.

$$L_{WKL} = \sum_{d_j \in \mathcal{D}^+} (1 - q_j)^{\gamma_1} p_j \ln p_j + \sum_{d_i \in \mathcal{D}^-} q_i^{\gamma_{2,i}} p_i \ln p_i$$
$$- \sum_{d_j \in \mathcal{D}^+} (1 - q_j)^{\gamma_1} p_j \ln q_j - \sum_{d_i \in \mathcal{D}^-} q_i^{\gamma_{2,i}} p_i \ln q_i$$
$$\geq \sum_{d_j \in \mathcal{D}^+} p_j \ln p_j + \sum_{d_i \in \mathcal{D}^-} p_i \ln p_i$$
$$- \sum_{d_j \in \mathcal{D}^+} \gamma_1(1 - q_j) p_j \ln q_j - \sum_{d_i \in \mathcal{D}^-} q_i^{\gamma_{2,i}} p_i \ln q_i$$
$$\geq r_1 \sum_{d_j \in \mathcal{D}^+} p_j \ln \frac{p_j}{q_j} - \sum_{d_i \in D^-} p_i q_i^{\gamma_{2,i}} \ln q_i + \frac{\gamma_1}{\log e} \sum_{d_j \in \mathcal{D}^+} q_j \log q_j$$
$$+ (1 - \gamma_1) \sum_{d_j \in \mathcal{D}^+} p_j \ln p_j + \sum_{d_i \in \mathcal{D}^-} p_i \ln p_i$$
□

The first component of the right hand side in Inequalities (3) and (4) is KL divergence for positive documents. The sum of the first and second components on the right-hand side approaches a constant lower bound, reached when $p_i = q_i$ for all positive documents and $q_i = 0$ for all negative documents. The third component is the negative entropy of positive documents. The third component is bounded by $-\frac{2\gamma}{e}$, approached when all $q_j$ values are equal for all positive documents $d_j$. This is shown in the theorem below.

**Theorem** 2. *Constant-bounded loss. If $\gamma_{2,i} > 0$, and when $\gamma_1 \geq 1$ or $\gamma_1 = 0$,*

$$L_{WKL} \geq \sum_{d_i \in \mathcal{D}^-} p_i(-1 + \ln p_i) - \frac{2\gamma_1}{e}. \tag{5}$$

*If $\gamma_{2,i} > 0$, and when $0 < \gamma_1 < 1$,*

$$L_{WKL} \geq \sum_{d_i \in \mathcal{D}^-} p_i(-\gamma_1 + \ln p_i) - \frac{2\gamma_1}{e}. \tag{6}$$

*If $\gamma_{2,i} = 0$, and when $\gamma_1 \geq 1$ or $\gamma_1 = 0$,*

$$L_{WKL} \geq -\frac{2\gamma_1}{e}. \tag{7}$$

*If $\gamma_{2,i} = 0$, and when $0 < \gamma_1 < 1$,*

$$L_{WKL} \geq (1 - \gamma_1) \sum_{d_j \in \mathcal{D}^+} (-p_j + p_j \ln p_j) - \frac{2\gamma_1}{e}. \tag{8}$$

Notice that $p_i$ and $p_j$ from the teacher's model in the above bound expressions are constants. The proof for Theorem 2 is based on Theorem 1 and is listed in Appendix A.

Based on the components of the derived lower bound, minimizing WKL will minimize the original KL-divergence loss for positive

documents and maximize the entropy among them. This lower bound minimization implies a balanced trend towards a narrower gap between teacher's and student's predictions of positive documents and relatively equal student predictions among them while preferring low scores for negative documents.

## 4 RELATIVE GRADIENT CONTRIBUTIONS

We analyze the impact of up-weighting and down-weighting individual KL-divergence terms in terms of their corresponding gradient contributions for parameter update during model refinement because gradients of the loss controls the update size to the network weight parameters in the SGD-based training process. Let $\theta$ be one of parameters $\Theta$ used in the computation network that maps the input features to score $S(Q, d_i, \Theta)$ for each document $d_i$. defined in Section 2. Then given Loss $L_A$, and $A$ can be $WKL$, $BKL$, or others.

$$\frac{\partial L_A}{\partial \theta} = \sum_{d_i \in \mathcal{D}^+ \cup \mathcal{D}^-} \frac{\partial L_A(i)}{\partial q_i} \frac{\partial q_i}{\partial S(Q, d_i, \Theta)} \frac{\partial S(Q, d_i, \Theta)}{\partial \theta}$$

where $L_A(i)$ is the relevant loss term contributed by document $d_i$.

For KL divergence loss $L_{KL}$ in Equation (1), $L_{KL}(i) = p_i \ln \frac{p_i}{q_i}$.

$$\frac{\partial L_{KL}(i)}{\partial q_i} = -\frac{p_i}{q_i}.$$

To understand if a loss function $L_A$ follows the KL divergence loss when a teacher model performs better than a student or not, we compare the pairwise ratio of the gradient contribution from document $d_i$ in above additive formulas for $\frac{\partial L_A(i)}{\partial q_i}$ compared to $\frac{\partial L_{KL}(i)}{\partial q_i}$. Namely

$$\frac{\partial L_A(i)}{\partial q_i} = g_A \frac{\partial L_{KL}(i)}{\partial q_i} \tag{9}$$

The top portion of Table 1 gives the expected behavior of a knowledge distillation loss compared to KL divergence loss when a teacher model performs better or worse than a student. The bottom of portion of Table 1 explains the meaning of different ranges of $g_A$ value on the gradient contribution of document $d_i$. Here the relative performance assessment of a teacher model and a student model for a document is defined below based on the relative ratio of teacher prediction and student prediction.

- A teach model performs better than a student model when $p_i > q_i$ if $d_i$ is a positive document when $p_i < q_i$ if $d_i$ is a negative document
- A teach model performs worse than a student model when $p_i < q_i$ if $d_i$ is a positive document, and when $p_i > q_i$ if $d_i$ is a negative document.

For $L_{WKL}$, the contribution $L_{WKL}(i)$ from document $d_i$ is $(1 - q_i)^{\gamma_1} p_i \ln \frac{p_i}{q_i}$ for a positive document, and $q_i^{\gamma_{2,i}} p_i \ln \frac{p_i}{q_i}$ for a negative document. It is easy to verify that

$$\frac{\partial L_{WKL}(i)}{\partial q_i} = g_{WKL} \frac{\partial L_{KL}(i)}{\partial q_i} \tag{10}$$

where

$$g_{WKL} = \begin{cases} (1 - q_i)^{\gamma_1 - 1} \times \left(\gamma_1 q_i \ln \frac{p_i}{q_i} + 1 - q_i\right) & \text{if } d_i \in \mathcal{D}^+; \\ q_i^{\gamma_{2,i}} \times \left(1 - \gamma_{2,i} \ln \frac{p_i}{q_i}\right) & \text{if } d_i \in \mathcal{D}^-. \end{cases}$$

| Scenarios | Expected behavior |
|---|---|
| Teacher is better than student | $g_A \geq 1$ preferred. At least $g_A > 0$ |
| Teacher is worse | $g_A \leq 0$ preferred. At most $g_A < 1$ |

| Condition | Behavior interpretation on $d_i$ contribution by $L_A$ |
|---|---|
| $g_A > 1$ | Aggressively follow KL divergence |
| $g_A = 1$ | Exactly follow KL divergence |
| $0 < g_A < 1$ | Conservatively follow |
| $g_A = 0$ | Not follow. No contribution from $d_i$ in $L_A$. |
| $g_A < 0$ | Not follow. Deviate from $d_i$ from KL Divergence |

Table 1: Expected gradient contribution behavior from document $d_i$ in loss $L_A$ compared to KL divergence

For KL divergence regularized together with the log-likelihood (Equation (2)),

$$g_{KLL} = \begin{cases} 1 + \frac{\lambda}{p_i} & \text{if } d_i \in \mathcal{D}^+; \\ 1 & \text{if } d_i \in \mathcal{D}^-. \end{cases}$$

Thus KLL always follows KL divergence loss even a teacher performs worse than a student. BKL in [45] improves this by combining the KL divergence with a log likelihood rank loss linearly using a small $\lambda$ parameter value.

$$g_{BKL} = \begin{cases} 1 - \frac{\lambda}{p_i} q_i \log(e \times q_i) & \text{if } d_i \in \mathcal{D}^+; \\ 1 - \frac{q_i \lambda}{p_i \ln 2} & \text{if } d_i \in \mathcal{D}^-. \end{cases}$$

Constant $e$ is the base of the natural logarithms.

Table 2 gives a comparison of the behavior of WKL and BKL for the gradient contribution of an individual document $d_i$ compared to KL divergence compared to KL divergence based on different $g_{WKL}$ and $g_{BKL}$ value ranges and the relative ratio of teacher's and student's predictions $\frac{p_i}{q_i}$. Notice that when $p_i > q_i$, we consider a teacher model *performs better* than a student if $d_i \in \mathcal{D}^+$, and *performs worse* if $d_i \in \mathcal{D}^-$. When $p_i < q_i$, we consider this teacher model *performs better* than a student if $d_i \in \mathcal{D}^-$, and *performs worse* if $d_i \in \mathcal{D}^+$.

Table 2 lists the conditions representing three significant misbehavior regions, to be illustrated in Figure 3(a), in which BKL fails to meet the expectation discussed in the top portion of Table 1. WKL is well-behaved as shown from this table and its behavior is formally characterized by the following theorem.

**Theorem** 3. *When a teacher model performs better than a student model in ranking a document for a query, $g_{WKL} > 0$. When this teacher model performs worse, $g_{WKL} < 1$, and $g_{WKL} \leq 0$ when $q_i \geq \max(e \times p_i, \frac{1}{\gamma_1 + 1})$ for $d_i \in \mathcal{D}^+$ and when $\frac{p_i}{q_i} \geq e^{\frac{1}{r_{2,i}}}$ for $d_i \in \mathcal{D}^-$.*

Proof. For $d_i \in \mathcal{D}^+$, we consider the ratio $g_{WKL}$ defined in Equation (10) in two cases.

- When $p_i > q_i$, $\gamma_1 q_i \ln \frac{p_i}{q_i} + 1 - q_i > 0$. Thus $g_{WKL} > 0$.
- When $p_i < q_i$, $\gamma_1 q_i \ln \frac{p_i}{q_i} + 1 - q_i < 1 - q_i$. Thus $g_{WKL} < (1-q_i)^{\gamma_1} \leq 1$.
  When $q_i \geq e \times p_i$, $\gamma_1 q_i \ln \frac{p_i}{q_i} + 1 - q_i \leq -\gamma_1 q_i + 1 - q_i \leq 0$ if $q_i \geq \frac{1}{\gamma_1 + 1}$.
  Thus $g_{WKL} \leq 0$ when $q_i \geq \max(e \times p_i, \frac{1}{\gamma_1 + 1})$.

For $d_i \in \mathcal{D}^-$,
- When $p_i > q_i$, $1 - \gamma_{2,i} \ln \frac{p_i}{q_i} < 1$. Then $g_{WKL} < q_i^{\gamma_{2,i}} \leq 1$.
  When $\frac{p_i}{q_i} \geq e^{\frac{1}{r_{2,i}}}$, $1 - \gamma_{2,i} \ln \frac{p_i}{q_i} \leq 0$. Then $g_{WKL} \leq 0$.
- When $p_i < q_i$, $1 - \gamma_{2,i} \ln \frac{p_i}{q_i} > 0$. Then $g_{WKL} > 0$.

$\square$

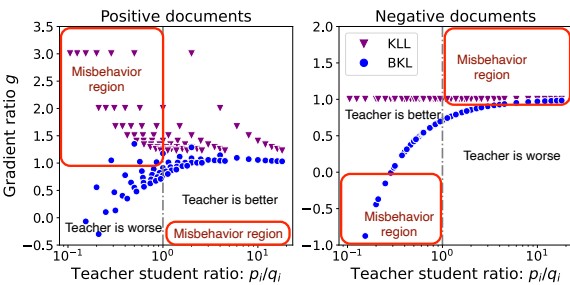

(a) Relative gradient contribution ratio $g$ of BKL and KLL

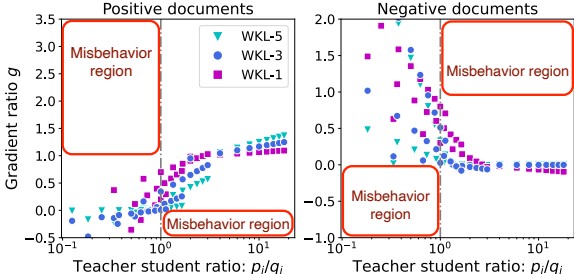

(b) Gradient contribution ratio of WKL with $\gamma_1 = \gamma_{2,i}$=5, 3, or 1

Figure 3: WKL vs. BKL & KLL when teacher is better or worse

To illustrate the comparison in Table 2 using an example, Figure 3(a) plots the gradient contribution ratio $g_{BKL}$ in a blue dot and $g_{KLL}$ in a purple triangle with $\lambda = 0.1$. The x-axis is $\frac{p_i}{q_i}$ varying from 0.01 from 10 at Figure 3(b) plots $g_{WKL}$ with $\gamma_1 = \gamma_{2,i}$=5, 3, or 1 marked as WKL-5 (a light blue triangle), WKL-3 (a dark blue dot), and WKL-1 (a purple square), respectively.

The rectangle red boxes marked the misbehavior regions show that the areas where the gradient contribution ratio values do not match the expected behavior described in the top portion of Table 1. From Figure 3(b), $g_{WKL}$ values are outside the red misbehavior regions, and thus WKL follows KL divergence loss if the teacher model does better than the student, but it restrains gradient update with a conservative size, or deviates in an opposite direction when the teacher is worse.

From Figure 3(a), there are three significant misbehavior regions in red into which BKL gradient ratios fall, meaning BKL fails to meet the expectations as summarized in Table 2. There are two misbehavior regions into which KLL falls.

For example, Figure 3(b) illustrates that for positive documents, when the teacher performs better with $\frac{p_i}{q_i} > 1$, $g_{WKL} > 0$ or exceeds 1 and WKL allows the student to follow the teacher's parameter

| Scenarios | Behavior of WKL | Behavior of BKL |
|---|---|---|
| **Positive document $d_i$** | | |
| Teacher: better | $g_{WKL} > 0$ | $g_{BKL} > 0$ |
| $p_i > q_i$ | Conservatively or aggressively follow | Conservatively or aggressively follow |
| Student: better | $g_{WKL} < 1$ to conservatively follow or deviate | $g_{BKL}$ varies from negative to positive |
| $p_i < q_i$ | $g_{WKL} \leq 0$ when $\frac{q_i}{p_i} \geq e, q_i \geq \frac{1}{r_1+1}$ to deviate | **Misbehavior**: $g_{BKL} > 1$ aggressively follows when $q_i > e^{-1}$ |
| **Negative document $d_i$** | | |
| Student: better | $g_{WKL} < 1$ to conservatively follow or deviate | **Misbehavior**: When $p_i >> q_i$, $g_{BKL} \approx 1$ |
| $p_i > q_i$ | $g_{WKL} \leq 0$ when $\frac{p_i}{q_i} \geq e^{\frac{1}{r_{2,i}}}$ to deviate | Otherwise $0 < g_{BKL} < 1$ to conservatively follow |
| Teacher: better | $g_{WKL} > 0$ | $0 < g_{BKL} < 1$ if $\frac{p_i}{q_i} > \frac{\lambda}{\ln 2}$ to conservatively follow |
| $p_i < q_i$ | Conservatively/aggressively follow | **Misbehavior**: $g_{BKL} \leq 0$ if $\frac{p_i}{q_i} \leq \frac{\lambda}{\ln 2}$ to deviate |

**Table 2: A comparison of relative gradient contributions by document $d_i$ in $L_{WKL}$ and $L_{BKL}$ compared to $L_{KL}$, $p_i$ is teacher prediction, $q_i$ is student prediction.**

update direction. When the teacher under-performs with $\frac{p_i}{q_i} < 1$, $g_{WKL}$ become close to 0 or even negative, and the student does not learn much from the teacher or its learning deviates from the teacher's learning direction. In comparison from the left portion of Figure 3(b), BKL still forces the student to follow the teacher's direction with $g_{BKL} > 0$ or even $> 1$ in most cases when the teacher is worse. Thus WKL's design corrects the misbehavior of BKL.

## 5 EVALUATION RESULTS

### 5.1 Evaluation setup for student models

We apply WKL in refining three student models during training.

- The SPLADE model [7, 9] which computes the weight score $w_j$ of $j$-th token term for a sparse vector of document $d$ as

$$w_j = \sum_{i \in d} log(1 + ReLU(H(h_i)^T E_j + b_j))$$

where document $d$ consists of a sequence of BERT last layer's embeddings $(h_1, h_2, \cdots, h_n)$. $E_j$ is the BERT input embedding of the $j$-th token and $b_j$ is a token level bias. $H(.)$ is a linear layer with activation and layer normalization.

- Two-stage search pipeline that combines the first-stage SPLADE retrieval and the second-stage ColBERT top-$k$ ranking. ColBERT's scoring formula is:

$$\sum_{h_i \in M(Q, \Theta)} max_{h_j \in M(d, \Theta)} H(h_i)^T H(h_j)$$

where each document $d$ and given query $Q$ use a multi-vector representation $M(d, \Theta)$ and $M(Q, \Theta)$ respectively, and $h_i, h_j$ are BERT last layer's embeddings and $H(.)$ is one linear layer with normalization on the output representation.

- Dense single-vector retriever SimLM [42]. It is a state-of-the-art dual-encoder with optimized pretraining [42, 43].

**WKL parameters.** We have considered a special version of WKL. For negative document $d_i$, we set $\gamma_{2,i} = \gamma_1 - \beta_i$. The exponent weight bias $\beta_i$ is defined as

$$\beta_i = \alpha \left( \frac{1}{\pi(i)} - \frac{1}{|\mathcal{D}^+|} \sum_{d_j \in \mathcal{D}^+} \frac{1}{\pi(j)} \right). \tag{11}$$

Here $\pi(i)$, $\pi(j)$ are the rank of negative document $d_i$ and positive document $d_j$ respectively. Bias $\beta_i$ represents the importance of correcting the ranking position of negative document $d_i$, compared against the harmonic average position of positive documents. The above use of a rank position is motivated by the previous work which considers the relevance gain by swapping two documents in a ranked order, e.g. LambdaMART [1] and CL-DRD [47]. The above expression satisfies $|\beta_i| < \alpha$. Among negative documents, if $q_i > q_j$, document $d_i$ is ranked before $d_j$. $\frac{1}{\pi(i)} > \frac{1}{\pi(j)}$. Thus $\beta_i > \beta_j$. Then $\gamma_{2,i} < \gamma_{2,j}$. That meets the requirement specified in Section 3.2.

Exponent bias $\beta_i$ is updated based on its rank position immediately after each training iteration where $q_i$ is recomputed, which makes the loss function non-differentiable. Thus during training, we opt to periodically update $\beta_i$ using the latest student's model performance, and the priority adjustment of each negative document is stable for a block of training iterations. This design allows $\beta_i$ to be treated as a constant in the loss function. This is a reasonable tradeoff as model refinement that addresses ranking accuracy for a negative document takes a number of iterations and continuous $\beta_i$ adjustment for such a document may not yield sufficient benefits.

Since $\gamma_1$ and $\alpha$ determine the value of $\gamma_{2,i}$ for every document $d_i$, the rest of this section will use two hyperparameters $\gamma_1$ and $\alpha$ to adjust the configuration of WKL, and investigate the sensitivity with different choices of $\gamma_1$ and $\alpha$ values in model refinement.

**Datasets and metrics.** We use the MS MARCO datasets for full passage ranking [2, 5]. MS MARCO contains 8.8 million passages and 502,940 training queries with binary judgment labels for each query. The development (Dev) query set contains 6980 test queries while the test sets in TREC deep learning (DL) 2019 and 2020 tracks provide 43 and 54 queries, respectively. Following the common practice, we report mean reciprocal rank (MRR@10) for the Dev set and NDCG@10 score for TREC DL test sets. The recall ratio at 1000 is another metric which is the percentage of relevant-labeled results appeared in the final top-1000 results.

The second data collection used is called BEIR which contains the 13 publicly available datasets [40] for evaluating the zero-shot performance of the trained models. BEIR is a heterogeneous benchmark containing a variety of IR tasks.

Our evaluation implementation uses C++ and Python. The implementation of SPLADE model follows its official release [39] and sparse retrieval code in PISA [29] with some optimization [18, 32]. We follow the SBERT library [36] to implement ColBERT. Two teachers are used during training. For SimLM, we use the code and checkpoint released in the SimLM project GitHub. The cross encoder teacher adopted for ColBERTv2 and SPLADE is MiniLM-l-6-v2 [30] with 0.407 MRR@10 on MS MARCO Dev on top of SPLADE retrieval. For SimLM, we use a cross encoder teacher from the released SimLM project [42] with 0.438 MRR@10. More information on training and configurations can be found in Appendix B.

## 5.2 Refinement of student models

**Two-stage search with ColBERT and SPLADE.** Table 3 compares the two-stage search trained under WKL and other distillation loss options in terms of MRR@10 or NDCG@10. The column for BEIR lists the average NDCG@10 across 13 datasets, and the detailed number on the zero-shot performance of two-stage SPLADE/ColBERT with WKL is in Table 4. We have listed published SPLADE++ and ColBERT performance. To demonstrate our evaluation is conducted competitive to the state-of-the-art research with multi-vector representations, this table lists dense retrievers with multi-vector representations like CITADEL and ALIGNER. We also include SLIM+ which improves multi-vector representations with a sparse scheme. The table also lists ColBERT re-ranking with uniCOIL first-stage retrieval and CQ quantization [46].

For WKL, re-ranking is applied to top 1,000 results of SPLADE retrieval. The middle portion of 3 also lists the results under different loss options including MarginMSE loss [15] and BKL [45]. "KLDiv_logL" uses $L_{KLL}$ with negative log likelihood loss on in-batch negatives plus KL-divergence loss. "CL-DRD" is a listwise loss in CL-DRD for curriculum learning [47]. Recall@1000 is the same as SPLADE for all loss options due to re-ranking and is not listed. Training for all loss options is conducted under the same training setup in terms of negative samples, the starting warm-up checkpoint, and the machine environment. WKL visibly outperforms other loss options for the test datasets.

Overall speaking, this table shows that the two-stage SPLADE and ColBERT search refined with WKL delivers a good and well-balanced performance across the tested datasets. We have performed paired t-tests on the 95% confidence level. We mark the results with '†' if a baseline result is in statistically significant degradation from WKL. We do not perform t-tests on DL'19 and DL'20 as these sets are relatively small.

The bottom portion of Table 3 lists the SPLADE/ColBERT performance refined with WKL for MS MARCO passage Dev set under different hyperparameter $\gamma_1$ and $\alpha$ values. When $\gamma_1$ is too small, WKL performance is similar as KL-divergence and when $\gamma_1$ becomes too big, the gradient will reduce quickly towards 0 and such a value is not preferred. Thus ($\gamma_1$ =5, $\alpha$= 1.0) is a good choice.

**Student model SPLADE.** When focusing on the first retrieval stage with SPLADE, Table 5 compares different loss options and mark the results with '†' if a baseline result is in statistically significant degradation from WKL. Recall@1000 for these losses is near identical as 0.983 for the Dev set, and thus it is not listed. WKL still

| | Dev | DL19 | DL20 | BEIR(Avg) |
|---|---|---|---|---|
| | MRR@10 | NDCG@10 | NDCG@10 | NDCG@10 |
| **Related performance numbers from other papers** | | | | |
| SPLADE++ [8] | 0.380 | 0.732 | – | 0.507 |
| ColBERTv2 | 0.397 | – | – | 0.499 |
| uniCOIL/ColBERTv2 | 0.387 | 0.746 | 0.726 | – |
| SLIM++ [20] | 0.404 | 0.714 | 0.697 | 0.490 |
| CITADEL [21] | 0.399 | 0.703 | 0.702 | 0.501 |
| ALIGNER [31] | 0.403 | – | – | 0.511 |
| **SPLADE + top-1000 ColBERT re-ranking** | | | | |
| KLDiv | 0.406$^\dagger$ | 0.716 | 0.719 | 0.489 |
| MarginMSE | 0.406$^\dagger$ | 0.704 | 0.710 | 0.503 |
| KLDiv_logL | 0.405$^\dagger$ | 0.711 | 0.699 | 0.499 |
| CL-DRD | 0.406$^\dagger$ | 0.700 | 0.693 | 0.497 |
| BKL | 0.407 | 0.716 | 0.736 | 0.506 |
| **WKL ($\gamma_1$=5, $\alpha$=1)** | **0.411** | **0.744** | **0.741** | **0.515** |
| $\gamma_1, \alpha$ | for WKL in other values | | | |
| 2,0, 0.0 | 0.404 | 0.716 | 0.709 | |
| 2.0, 0.5 | 0.404 | 0.733 | 0.725 | |
| 2.0, 1.0 | 0.404 | 0.740 | 0.728 | |
| 3.0, 0.0 | 0.405 | 0.735 | 0.717 | |
| 4.0, 0.0 | 0.408 | 0.740 | 0.734 | |
| 4.0, 1.0 | 0.410 | 0.735 | 0.731 | |
| 4.0, 1.5 | 0.404 | 0.724 | 0.740 | |
| 5.0, 0.0 | 0.409 | 0.737 | 0.722 | |
| 5.0, 1.5 | 0.407 | 0.742 | 0.731 | |
| 6.0, 0.0 | 0.410 | **0.750** | 0.724 | |

**Table 3: Two-stage search with different loss options and WKL parameters**

| Dataset | BM25 | SPLADE++ | SimLM | ColBERTv2 | BM25/miniLM | WKL |
|---|---|---|---|---|---|---|
| Search Tasks | | | | | | |
| DBPedia | 0.313 | 0.436 | 0.351 | 0.446 | 0.400 | **0.459** |
| FiQA | 0.236 | 0.349 | 0.298 | 0.356 | 0.309 | 0.372 |
| NQ | 0.329 | 0.533 | 0.502 | 0.562 | 0.453 | **0.562** |
| HotpotQA | 0.603 | 0.693 | 0.568 | 0.667 | 0.677 | 0.692 |
| NFCorpus | 0.325 | 0.345 | 0.318 | 0.338 | **0.364** | 0.348 |
| T-COVID | 0.656 | 0.725 | 0.515 | 0.738 | **0.766** | 0.746 |
| Touche-2020 | **0.367** | 0.242 | 0.292 | 0.263 | 0.314 | 0.316 |
| Semantic Relatedness Tasks | | | | | | |
| ArguAna | 0.315 | 0.518 | 0.376 | 0.463 | 0.473 | **0.578** |
| C-FEVER | 0.213 | 0.237 | 0.171 | 0.176 | **0.239** | 0.231 |
| FEVER | 0.753 | 0.796 | 0.689 | 0.780 | 0.756 | 0.779 |
| Quora | 0.789 | 0.849 | 0.797 | **0.852** | 0.843 | 0.746 |
| SCIDOCS | 0.158 | 0.161 | 0.137 | 0.154 | 0.170 | 0.164 |
| SciFact | 0.665 | 0.710 | 0.559 | 0.568 | 0.697 | 0.698 |
| **Average** | 0.440 | 0.507 | 0.429 | 0.499 | 0.497 | **0.515** |
| **BM25 Diff** | – | 15.24% | -2.60% | 13.47% | 12.94% | **16.92%** |

**Table 4: Zero-shot performance (average NDCG@10) on BEIR**

outperforms other loss options with a smaller advantage in the Dev set while having a larger improvement of DL'19 and DL'20 test sets.

The bottom portion of Table 5 lists SPLADE performance refined with WKL under different hyperparameters $\gamma_1$ and $\alpha$ values. As one can see, $\gamma_1$ = 5 and $\alpha$ = 1 perform decently well.

**Student dense retrieval model SimLM**. WKL is applied to train on a SOTA dense retrieval model SimLM [42] and Table 6 lists MRR@10 and Recall@1000 for the Dev set, and NDCG for DL 19 and DL 20. WKL delivers 0.394 in MRR@10 with a warmup using KL divergence. Without warmup, WKL delivers 0.381. For dense retrievers, the released SimLM checkpoint [3] gives 0.344 MRR@10 using the standard MS MARCO. This is below 0.411 reported in [3]

| Loss options | Dev MRR@10 | DL19 NDCG@10 | DL20 NDCG@10 |
|---|---|---|---|
| KLDiv | $0.399^{\dagger}$ | 0.656 | 0.689 |
| MarginMSE | $0.397^{\dagger}$ | 0.664 | 0.678 |
| KLDiv_logL | $0.396^{\dagger}$ | 0.669 | 0.672 |
| CL-DRD | 0.400 | 0.674 | 0.662 |
| **WKL ($\gamma_1$=5, $\alpha$=1)** | **0.4013** | **0.7445** | **0.7206** |
| $\gamma_1, \alpha$ | **for WKL in other values** | | |
| 2,0, 0.0 | 0.3993 | 0.7435 | 0.7177 |
| 3.0, 0.0 | 0.4008 | 0.7348 | 0.7255 |
| 4.0, 0.0 | 0.4006 | 0.7309 | 0.7215 |
| 5.0, 0.0 | 0.4007 | **0.7456** | 0.7256 |
| 5.0, 1.5 | 0.4008 | 0.7192 | **0.7335** |
| 6.0, 0.0 | 0.4003 | 0.7317 | 0.7180 |

**Table 5: SPLADE with different losses and WKL parameters**

| Model | Dev MRR@10 | Dev R@1K | DL19 NDCG@10 | DL20 NDCG@10 |
|---|---|---|---|---|
| SimLM with title [42] | 0.4111 | 0.987 | 0.712 | 0.697 |
| SimLM w/o title | 0.344 | 0.947 | 0.650 | 0.641 |
| **Model refinement without title annotation** | | | | |
| KLDiv | 0.365 | 0.951 | 0.685 | 0.611 |
| WKL | 0.381 | 0.981 | 0.690 | 0.696 |
| **KL+WKL ($\gamma_1$=1,$\alpha$=0)** | **0.395** | **0.982** | **0.708** | **0.706** |
| $\gamma_1, \alpha$ | **for KL+WKL in other values** | | | |
| 2,0 | 0.394 | 0.980 | | |
| 3,0 | 0.393 | 0.981 | | |
| 4,0 | 0.392 | 0.981 | | |

**Table 6: Dense retriever SimLM with WKL and different parameters**

which evaluates the modified MS MARCO dataset with title annotation. Title annotation is considered unfair in [19] since the original dataset released doesn't utilize title information. The numbers reported from recent papers RocketQAv2 [33], LexMAE [37], RetroMAE and RetroMAE-2 [26, 43] were boosted by this title annotation. All experiments for WKL follow the standard approach of using the original MS MARCO without title annotation, and the WKL improvement in refining SimLM is reasonable compared to KL divergence loss.

The bottom portion of Table 6 lists the performance of SimLM refined with WKL after KL divergence warmup for MS MARCO passage Dev set under different hyperparameter $\gamma_1$ and $\alpha$ values. The result shows that ($\gamma_1$ =1, $\alpha$= 0) is a good choice for SimLM.

## 6 CONCLUDING REMARKS

The contribution of this work is to provide a detailed analysis of a generalized KL divergence loss in an easy-to-implement weighted format. Our lower bound analysis gives an insight into the behavior characteristic of WKL during model refinement. The relative gradient contribution study reveals that WKL follows the gradient update direction of KL divergence loss when the teacher model performs better than the student model for a training query. When this teacher performs worse, WKL deviates in an opposite update direction or restrains the update size cautiously in the same update

direction. Such adaptive learning behavior for knowledge distillation is accomplished by prioritizing scoring alignment of teacher and student models through KL divergence term weighting based on their relative performance for a positive or negative document. The evaluation gives evidences that WKL can effectively refine three student models for MS MARCO passages and BEIR datasets.

Our future work is to investigate the use of WKL in more ranking models and experiments. The limitation of this work is that the applicability of WKL is restricted to ranking applications where binary positive and negative labels are assigned per training queries. This considers that it is hard and costly to build a dataset at a large scale for ranker training with multi-level labels in practice. It is interesting to extend WKL in the future for training data with mult-level labels.

## A PROOF OF THEOREM 2

PROOF. let $RHS(i)$ be the $i$-th component of in the right-hand side of Inequality (3) or Inequality (4) in Theorem 1.

We further four cases in order to derive a lower constant bound.

**Case 1)** We consider the cases of $\gamma_{2,i} > 0$.

- We apply a known inequality: $\ln x \leq x - 1$ when $x$ is positive and the equality is reached when $x = 1$.

$$\sum_{d_j \in \mathcal{D}^+} p_j \ln \frac{p_j}{q_j} \geq \sum_{d_j \in \mathcal{D}^+} p_j(1 - \frac{q_j}{p_j}) \geq \sum_{d_j \in \mathcal{D}^+} p_j - \sum_{d_j \in \mathcal{D}^+} q_j.$$

The lower bound is accomplished when $p_j = q_j$ for all positive documents. Since

$$\sum_{d_j \in \mathcal{D}^+} p_j + \sum_{d_i \in \mathcal{D}^-} p_i = 1 \text{ and } \sum_{d_j \in \mathcal{D}^+} q_j + \sum_{d_i \in \mathcal{D}^-} q_i = 1,$$

$$\sum_{d_j \in \mathcal{D}^+} p_j \ln \frac{p_j}{q_j} \geq - \sum_{d_i \in \mathcal{D}^-} p_i + \sum_{d_i \in \mathcal{D}^-} q_i.$$

- Based on the above derivation, when $\gamma_1 \geq 1$ or $\gamma_1 = 0$,

$$RHS(1) + RHS(2) \geq - \sum_{d_i \in \mathcal{D}^-} p_i + \sum_{d_i \in \mathcal{D}^-} q_i - \sum_{d_i \in \mathcal{D}^-} p_i q_i^{\gamma_{2,i}} \ln q_i,$$

$$\geq - \sum_{d_i \in \mathcal{D}^-} p_i + \sum_{d_i \in \mathcal{D}^-} p_i(q_i - q_i^{\gamma_{2,i}} \ln q_i)$$

$$\geq - \sum_{d_i \in \mathcal{D}^-} p_i.$$

When $0 < \gamma_1 < 1$,

$$RHS(1) + RHS(2) = \gamma_1(- \sum_{d_i \in \mathcal{D}^-} p_i + \sum_{d_i \in \mathcal{D}^-} q_i) - \sum_{d_i \in \mathcal{D}^-} p_i q_i^{\gamma_{2,i}} \ln q_i,$$

$$\geq -\gamma_1 \sum_{d_i \in \mathcal{D}^-} p_i + \sum_{d_i \in \mathcal{D}^-} p_i(\gamma_1 q_i - q_i^{\gamma_{2,i}} \ln q_i)$$

$$\geq -\gamma_1 \sum_{d_i \in \mathcal{D}^-} p_i.$$

Notice that in the above derivation, Expression $q_i - q_i^{\gamma_{2,i}} \ln q_i$ has its lower bound achieved when $q_i$ is approaching 0. When $0 < \gamma_1 < 1$ Expression $\gamma_1 q_i - q_i^{\gamma_{2,i}} \ln q_i$ also has its lower bound achieved when $q_i$ is approaching 0.

- Now we derive a lower bound for $RHS(3) = \frac{\gamma}{\log e} \sum_{d_i \in \mathcal{D}^+} q_i \log q_i$. Since function $x \log x$ is convex and following Jensen's inequality on a convex function,

$$\frac{\sum_{d_j \in \mathcal{D}^+} q_j \log q_j}{s} \geq (\frac{\sum_{d_j \in \mathcal{D}^+} q_j}{s}) \log(\frac{\sum_{d_j \in \mathcal{D}^+} q_j}{s})$$

where $s = |\mathcal{D}^+|$. Let $z = \sum_{d_i \in \mathcal{D}^-} q_i$. Then

$$\sum_{d_j \in \mathcal{D}^+} q_j \log q_j \geq (1 - z) \log(\frac{1 - z}{s}) \geq -\frac{2 \log e}{e}.$$

Expression $(1 - z) \log(\frac{1-z}{s})$ is bounded by $-\frac{2 \log e}{e}$ by computing its minimum value.

Adding the above component lower bounds together. When $\gamma_1 \geq 1$ or $\gamma_1 = 0$,

$$L_{WKL} = RHS(1) + RHS(2) + RH(3) + RHS(4) \geq \sum_{d_i \in \mathcal{D}^-} p_i(-1 + \ln p_i) - \frac{2\gamma_1}{e}.$$

When $0 < \gamma_1 < 1$,

$$L_{WKL} = RHS(1) + RHS(2) + RH(3) + RHS(4) + RHS(5)$$
$$\geq (1 - \gamma_1) \sum_{d_j \in \mathcal{D}^+} p_i \ln p_i + \sum_{d_i \in \mathcal{D}^-} p_i(-\gamma_1 + \ln p_i) - \frac{2\gamma_1}{e}$$

**Case 2)** We consider the case of $\gamma_{2,i} = 0$ and $\gamma_1 = 0$. In this case, WKL is the same as KL divergence loss.

$$L_{WKL} = L_{KL} \geq 0.$$

The lower bound is accomplished $p_i = q_i$ for all positive and negative documents $d_i$.

**Case 3)** Now we consider the cases of $\gamma_{2,i} = 0$, and $\gamma_1 > 0$. There are two subcases.

**Subcase 3.1)** When $\gamma_1 > 1$, from Inequality (3),

$$L_{WKL} \geq \sum_{d_j \in \mathcal{D}^+} p_j \ln \frac{p_j}{q_j} - \sum_{d_i \in \mathcal{D}^-} p_i \ln q_i$$
$$+ \frac{\gamma_1}{\log e} \sum_{d_j \in \mathcal{D}^+} q_j \log q_j + \sum_{d_i \in \mathcal{D}^-} p_i \ln p_i$$
$$\geq \frac{\gamma_1}{\log e} \sum_{d_j \in \mathcal{D}^+} q_j \log q_j$$
$$\geq -\frac{2\gamma_1}{e}.$$

**Subcase 3.2)** When $0 < \gamma_1 < 1$, from Inequality (4),

$$L_{WKL} \geq r_1 \sum_{d_j \in \mathcal{D}^+} p_j \ln \frac{p_j}{q_j} - \sum_{d_i \in \mathcal{D}^-} p_i \ln q_i + \frac{\gamma_1}{\log e} \sum_{d_j \in \mathcal{D}^+} q_j \log q_j$$
$$+ (1 - \gamma_1) \sum_{d_j \in \mathcal{D}^+} p_j \ln p_j + \sum_{d_i \in \mathcal{D}^-} p_i \ln p_i$$
$$\geq r_1 \sum_{d_j \in \mathcal{D}^+} p_j \ln \frac{p_j}{q_j} + \sum_{d_i \in \mathcal{D}^-} p_i \ln \frac{p_i}{q_i}$$
$$+ \frac{\gamma_1}{\log e} \sum_{d_j \in \mathcal{D}^+} q_j \log q_j + (1 - \gamma_1) \sum_{d_j \in \mathcal{D}^+} p_j \ln p_j$$
$$\geq \gamma_1(- \sum_{d_i \in \mathcal{D}^-} p_i + \sum_{d_i \in \mathcal{D}^-} q_i) + (\sum_{d_i \in \mathcal{D}^-} p_i - \sum_{d_i \in \mathcal{D}^-} q_i)$$
$$- \frac{2\gamma_1}{e} + (1 - \gamma_1) \sum_{d_j \in \mathcal{D}^+} p_j \ln p_j$$
$$\geq (1 - \gamma_1) \sum_{d_i \in \mathcal{D}^-} p_i - (1 - \gamma_1) - \frac{2\gamma_1}{e} + (1 - \gamma_1) \sum_{d_j \in \mathcal{D}^+} p_j \ln p_j$$
$$= (1 - \gamma_1) \sum_{d_j \in \mathcal{D}^+} (-p_j + p_j \ln p_j) - \frac{2\gamma_1}{e}.$$

$\square$

## B  TRAINING STEPS AND CONFIGURATIONS

Training for each student model involves two steps: Step 1 is to warm up the student model with knowledge distillation following a fixed teacher model. Step 2 is to use the proposed WKL loss or other loss options to refine the student retriever model and the student re-ranker model separately. When we compare different loss functions for the refinement, we always start from the same model after warm-up and refine it using the same set of training triplets and the same teacher model. In this way, we rule out the potential influence caused by different implementation details in performance comparison.

The cross encoder teacher adopted for ColBERTv2 and SPLADE is MiniLM-l-6-v2 [30]. For SimLM, we use a cross encoder from the released SimLM project [42]. Following the setting of SPLADE++, we use co-Condenser [4] as the pretrained starting checkpoint and adopt sentenceBERT [13] as the ranker to select hard negatives. This warm-up step chooses margin-MSE [15] as a loss for knowledge distillation for both retriever and re-ranker. To train the retrieval model, we also add additional sparsity regularization with coefficients 0.008 and 0.01 for a query and documents respectively, following SPLADE++. This observation aligns with the results reported in TAS-B [16]. The above warm-up step allows the SPLADE retriever to reach 0.394 MRR@10 and the ColBERT re-ranker to deliver 0.399 MRR@10.

In Step 2 for model refinement, we use the WKL loss for knowledge distillation or another loss function to compare. We index the corpus with a warm-up retrieval model using PISA [29]. To speedup training, we only retrieve the top 100 documents (passages for MS MARCO) per query for re-ranking during training. Negative sampling uses the top 20 documents per training query after re-ranking as candidate hard negatives. During model refinement, we sample negative examples from these 20 documents so that the total number of positive and negative documents is a fixed constant, limited by the available GPU memory. For our machine environment, this fixed constant is 6. Namely, if there are 2 positive documents for a query, we sample at most 4 negative documents.

In terms of training machine resources and parameters, we use four NVIDIA V100 GPUs to warm up and refine SPLADE with the training batch size as 128 queries and to warm up and refine ColBERT with a batch size of 32 queries. This training resource usage is reasonable compared to what has been used in the previous work [16, 33, 34]. Learning rates 2e-5 and 1e-5 are used in the warm-up step and the refinement step, respectively. We update the exponent weight bias $\beta_i$ discussed above every 2000 training batches, as more frequent update does not lead to an improvement. When training the student retriever, to avoid the expensive re-indexing time during this update, we re-evaluate the top 50 documents per training query as an approximation using the model checkpoint saved after every 2000 batches. The above refinement with WKL for training takes less than 20 epochs to converge.

The selected default WKL parameters are $(\gamma, \alpha) = (5, 1)$ for ColBERT and SPLADE, and $(\gamma, \alpha) = (1, 0)$ for SimLM. Section 5 examines the choices and sensitivities of these WKL parameters for these models.

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
