# OpenReview forum: "On Adaptive Knowledge Distillation with Generalized KL-Divergence Loss for Ranking Model Refinement"
_ACM.org/SIGIR/ICTIR/2024/Conference — ICTIR 2024_

### Official Review · Reviewer_RDer · 2024-05-09

**Rating:** -1
**Confidence:** 4

**Objective Part Of Review:**

The paper is well motivated and written. A detailed analysis of the proposed approach is also included.

**Subjective Part Of Review:**

Although the proposed approach is clear and well motivated, the experiment can be better organized to make the advantage of the approach more clear to readers (See below).

The evaluation in this paper seems really strange to me. (1) The baseline uses uniCOIL with ColBERTv2 reranking while the authors choose SPLADE + ColBERT re-ranking. Why not use SPLADE++ with ColBERTv2 reranking as comparison? I believe both models are distilled with a simple strategy (e.g., KLDiv or MarginMSE).  (2) Why not directly report the end-to-end retrieval performance for the WKL distilled SPLADE and ColBERT in Table 3? Using multi-stage pipeline with WKL distilled SPLADE and ColBERT hides many information behind. In my opinion, the experiments can be organized in a much better way. For example, the main experiment (table 3) just reports the results on the proposed methods (and other distillation variants) on SPLADE, SimLM and ColBERT. Comparing with other state-of-the-art can be in another table, which is not the key message.

The other concern to me is the motivation part. The paper mention that KL-divergence loss does not differentiate positive and negative documents for a training query. Although I agree that the KL-divergence approach may over optimize or mimic teachers’ behavior, using the positive from the existing training data (e.g. MS MARCO) may also suffer from false positive. That is, pervious work has highlighted many top retrieved passages (which are not labeled) are actually relevant to the queries. Thus, MS MARCO dataset may not be perfect suite to demonstrate the advantage of the method. In addition, the proposed method still uses MSE margin as warm up training. But why not directly use the proposed approach from the beginning? It seems to me that the major gain from proposed approach is that it fuse the signals from margin MSE and WKL.

---

### Official Review · Reviewer_ro2g · 2024-05-15

**Rating:** 1
**Confidence:** 3

**Objective Part Of Review:**

The paper deals with Knowledge Distillation for training a student neural document ranking model from a teacher, with the additional challenge that the teacher may not perform well in all cases. The paper proposes weighted KL-Divergence (WKL), a generalised KL-divergence loss formula. It also provides a detailed analysis of its theoretical properties, including a lower bound analysis and analysis of gradient contributions. Evaluation experiments compare the proposed approach to state-of-the-art approaches on the MS MARCO dataset. Limitations of the work are discussed at the end.

There are a few issues that should be cleared up in the final paper:
- For the evaluation, suddenly a "special version of WKL" is introduced without any further details. What is special about it? Why is this special version used? Why is it suddenly introduced in the evaluation section and not as part of the model presentation?
- In the experiments, two teachers are used, with different student models being paired with specific teacher models. The reason for this specific pairing of teacher and student models should be explained. Why can a single teacher model not be used for all student models?
- "warmup" is suddenly mentioned in Section 5.2 without being explained.

**Subjective Part Of Review:**

I like the theoretical approach of the paper, although I didn't get a chance to look in detail at the theory - it's good to see reasons for why a model behaves as it does. I'm sure that other researchers will gain insights from this approach.

---

### Official Review · Reviewer_TLEw · 2024-05-16

**Rating:** 1
**Confidence:** 4

**Objective Part Of Review:**

* Is the problem clearly stated?

  Yes.

  * Are the methods clearly described?

  The methods are described clearly for the most part. Some minor changes would improve clarity.
  - Figure 2, state whether this is a teacher or student ranking. Describe what the red and black circles mean (textured circles would be nicer for colorblind readers). Adjust the upweight arrow so that it is clearly pointing at the red document on the right. Describe the axes (is Y the score and X the rank)?
  - The last paragraph of Section 3.1 is difficult to understand.
  - The paper uses log and ln interchangeably. Pick one.
  - Table 4, the caption should identify the configuration used to produce the results. A careful reader can figure it out, but why make them work?

  * Are the results clearly stated?

  Yes.

  * Are the various claims in the paper supported?

  Yes, although the support would be stronger if statistical significance tests had been used.

  * Is each concept and notation properly defined before it is used?

  Yes.

  * Can the abstract and the introduction be understood before one has read and understood the rest of the paper?

  Yes.

  * Did you spot any contradictions or other signs that something is wrong?

  No.

  * Is work that is directly relevant or even competing with the work in this paper cited?

  Yes.

**Subjective Part Of Review:**

* Did you find the paper easy to read and understand?

  The problem formulation and solution are clear for a general reader. The derivations and proofs are more challenging; most readers will skip over them.

  * Do you find the problem relevant?

  The problem is relevant. This new loss function provides a nice improvement on some datasets, and a small improvement on others.

  * Do you find the methods original?

  The method is original and worthwhile, but a small delta over prior work. It continues a recent trend of weighting training instances to focus on instances that are more informative in some way.

  * Do you find the results interesting?

  The results are moderately interesting.

  * Do you think that others in the ICTIR community will be interested in this work?

  Others may try out this loss function to see how it performs in their settings.

---

### Official Review · Reviewer_VpJq · 2024-05-17

**Rating:** 2
**Confidence:** 4

**Objective Part Of Review:**

This paper is very well-written and organized. The paper explains when knowledge distillation with BKL methods will cause wrong and undesired gradient updates, and proposes a generalized KLD loss (WKL) for knowledge distillation. The motivations and problem descriptions are clear. Furthermore, the paper provides theoretical proofs on WKL having a constant lower bound and demonstrates the gradient contributions with WKL fixes mis-behaviours by BKL. The evaluation setup considers three pipeline including SPLADE, SPLADE+ColBERT, and SimLM, and the evaluation results indicate WKL effectiveness. One thing that sounds a bit unexpected is that although the proposed loss WKL outperforms BKL and it is shown theoretically that addresses the issues with BKL, the improvements in some cases are less pronounced in table 3 and not statistically significant. Overall, I recommend the acceptance of this paper.

**Subjective Part Of Review:**

This paper very well aligns with ICTIR and is very well-presented. The paper identifies some issues when using BKL for knowledge distillation, illustrates them clearly and provides a solution (WKL) with theoretical analysis. The results indicate significant improvement from WKL over many of the baselines.

---

### Meta-Review · Area_Chair_pwKT · 2024-06-03

**Recommendation:** Accept (Oral)
**Confidence:** 5

**Metareview:**

The paper aims to improve distillation for training student ranking models from a teacher. The motivation is that the teacher is not always correct, and using KL-divergence loss will solely mimics the teacher while ignoring the ground-truth labels. The paper proposes weighted KL-Divergence Loss (WKL), which takes into account ground-truth labels. The paper also provides theoretical analysis of the proposed loss function.

In general this is a solid paper, with clear motivation and good empirical results.

Concerns:
1) Baselines should include more recent state-of-the-arts, e.g., SPLADE++ and ColBERTv2
2) The pipeline is somewhat complex, involving multiple teachers and training stages. Experiments section should be made more clear, and these complexities need to be justified through ablation.
3) Related to 2), it is important to highlight why the gains come from the new loss, not other elements in the training recipe such as multiple teachers or multi-stage training.